# PoTable: Programming on Tables to Reason Like a Distinguished Human Data Analyst

## Abstract

Table-based reasoning has garnered substantial research interest, particularly in its integration with Large Language Model (LLM) which has revolutionized the general reasoning paradigm. Numerous LLM-based studies introduce symbolic tools (*e.g.*, databases, Python) as assistants in complex information understanding and arithmetic computations. However, they emphasize extensive and flexible utilization of symbolic tools, without fully considering the intrinsic logic of the reasoning process. In this study, we propose PoTable as a simple yet effective table-based reasoning method. Specifically, PoTable features a planning phase and an executing phase, implemented with an LLM-based operation planner and code generator and a Python interpreter as the real-time executor. To incorporate logical top-level guidance, we split the entire reasoning process into several distinct analysis stages with macroscopic instruction injection. As the reasoning process is structured suitably under the top-level guidance with precise and specific goals, PoTable produces superior reasoning results with highly accurate, steply commented and completely executable code. To summarize, PoTable enjoys the advantages of accuracy and explainability that make it a distinguished tabular data analyst. Extensive experiments over three evaluation datasets from two public benchmarks on two backbones demonstrate the outstanding performance of PoTable. In particular, GPT-based PoTable achieves over 4% higher absolute accuracy than runner-ups on all evaluation datasets. Our code is available at https://anonymous.4open.science/r/PoTable-6788.

## 1 Introduction

Tables are widely applied in various scenarios (*e.g.*, healthcare (Ghasemi & Amyot, 2016), finance (Li et al., 2021)), since they can visually present the core information in various types of scientific documents (*e.g.*, articles, reports, websites) (Embley et al., 2006) through a structured format. With the growing development of AI techniques, there has been an increasing demand for automated table processing, attracting significant attention from both academia and industry (Borisov et al., 2022). Recently, the evolution of Large Language Model (LLM) (Zhao et al., 2023) has raised a brand new prompting paradigm for table processing (Lu et al., 2024). This training-free method facilitates complex understanding and reasoning procedures in table question answering (Pasupat & Liang, 2015), table fact verification (Chen et al., 2020) and other downstream tasks (shown in Figure 1(a)).

Throughout the history of humankind, tools have been regarded as the crystallization of human wisdom and a core factor in social productivity development (Washburn, 1960). This consensus has inspired LLM-based techniques to go a step further in simulating more extensive human behavior, *i.e.*, collaborating with symbolic tools to overcome LLMs' inherent limitations (Qu et al., 2024). In table processing, two unique challenges have been issued in earlier studies (Lu et al., 2024; Dong & Wang, 2024): (1) Tables are structured in two-dimension, leading to unstable memorization of LLMs trained in next-token prediction mode (Sui et al., 2024). (2) Table-based reasoning inevitably involves logical and arithmetic operations, and LLMs may produce misleading results due to their limited calculation abilities. Nevertheless, with a rising trend to utilize databases (Li et al., 2023b), Python (Chen et al., 2022; Gao et al., 2023) and other symbolic tools as assistants, recent approaches effectively reduce table processing errors and misleading computational results by storing the tabular data into internal structure types (*e.g.*, arrays, database tables) and executing syntactic computation commands (*e.g.*, SQL, Python code) (Cheng et al., 2023; Cao et al., 2023; Nahid & Rafiei, 2024).

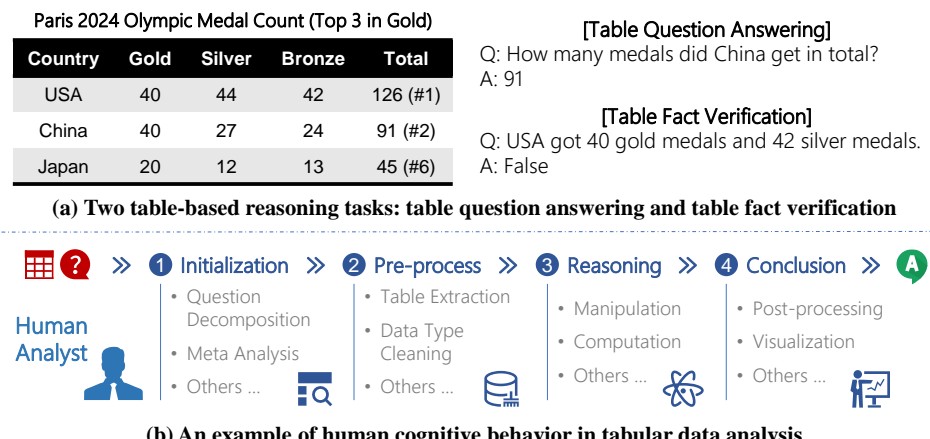

(a) Two table-based reasoning tasks: table question answering and table fact verification

(b) An example of human cognitive behavior in tabular data analysis

Figure 1: Illustration of (a) two table-based reasoning tasks evaluated in our study, and (b) the human analyst follows top-level logical guidance to plan and execute operations under distinct stages to produce highly accurate and explainable answers.

Despite their promising results in some scenarios, they emphasize extensive and flexible utilization of symbolic tools without fully considering *the intrinsic logic of the overall reasoning process*. Some earlier studies promote LLMs to generate complete task programs without intermediate decomposition (Cheng et al., 2023; Cao et al., 2023). Recent studies adopt dynamic state observation and autonomous operation planning, lacking explicit global guidance during reasoning (Wang et al., 2024b; Zhang et al., 2024b). These approaches may encounter missing steps or misleading details when handling complex tasks with numerous reasoning operations, leading to sub-optimal results. Moreover, verifying the accuracy of these reasoning processes can be time-consuming. For instance, it is hard to quickly judge whether an existing operation is needed immediately or can be deferred.

Along this line, in tool-based reasoning, integrating **logical top-level guidance** is necessary to reduce the possibility of misleading chain steps while enhancing the explainability of the process. Some studies design instructions only for special operations (*e.g.*, table decomposition) (Ye et al., 2023; Nahid & Rafiei, 2024), yet the overall top-level guidance integration remains underexplored. Metaphorically speaking, existing methods behave more like a junior student rather than a distinguished human data analyst (shown in Figure 1(b)), which follows a relatively standard top-level stage-split guidance in tabular mining and reasoning (Fayyad et al., 1996; Mariscal et al., 2010).

In this paper, we propose POTABLE (**P**rogramming **o**n **Table**s) as a simple yet effective table-based reasoning method. Inspired by plan-and-solve prompting (Wang et al., 2023), POTABLE features a planning phase for the operation chain production and an executing phase for code generation and real-time execution and feedback. Specifically, POTABLE integrates an LLM-based planner and code generator with a Python interpreter as the executor. To incorporate logical top-level guidance, we naturally split the entire process into several logical analysis stages with macroscopic instruction injection. At each stage, we set a general sub-goal and expect it to plan and execute the operations sequentially. Consequently, POTABLE produces superior reasoning results with highly accurate, steply commented and completely executable code.

POTABLE enjoys two advantages that make it a distinguished data analyst. **(1) Accuracy**: POTABLE can easily plan coherent operation chains under precise and specific sub-goals with less possibility of misleading or missing steps, producing more accurate results. **(2) Explainability**: POTABLE follows suitable structured top-level guidance with full operation code execution, making it easier to verify the completeness and accuracy of the reasoning process. Finally, we conduct extensive experiments over three evaluation datasets from two public benchmarks of table-based reasoning tasks on two backbones. POTABLE achieves more outstanding accuracy results than all LLM-based baselines. In particular, GPT-based POTABLE achieves over 4% higher absolute accuracy than runner-ups on all evaluation datasets. All experimental results and analyses validate the strong effectiveness of POTABLE. In summary, our main contributions can be listed as follows:

Figure 2: Illustration of our propose POTABLE, a simple yet effective table-based reasoning method. POTABLE follows logical top-level guidance as distinct analysis stage split: initialization, row selection, data type cleaning, reasoning and final answering. Each stage contains a planning phase to generate operation chains, and an executing phase to generate step code for real-time execution.

- We propose POTABLE, a simple yet effective table-based reasoning method that consists of an LLM and a Python interpreter to implement the planning and executing phases, producing highly accurate, steply commented and completely executable programs.

- We integrate logical top-level guidance into POTABLE by splitting the entire process into several logical analysis stages. By structuring the overall reasoning process in a suitable manner, POTABLE enjoys the advantages of high accuracy and explainability, making it behave like a distinguished human data analyst.

- Experimental results over three evaluation datasets from two public benchmarks of table-based reasoning tasks show the outstanding performance of POTABLE.

## 2 POTABLE

### 2.1 TASK FORMULATION

Our study focuses on two table-based reasoning tasks, *i.e.*, table question answering and table fact verification. Each sample can be represented as $(T, Q, A)$, where $T$ denotes the structured table, $Q$ denotes a question to be answered or a statement to be verified. Given $T$ and $Q$, our goal is to find the answer $A$ in the table question answering task, while in the table fact verification task, we have to decide $A = 1$ or $A = 0$ indicating whether the statement is true or false, respectively.

### 2.2 OVERVIEW

We propose POTABLE (**P**rogramming **o**n **Table**s), a simple yet effective table-based reasoning method shown in Figure 2. Specifically, POTABLE features a planning phase for the operation chain production and an execution phase for code generation and real-time execution and feedback, which is implemented by an LLM and a Python interpreter. POTABLE follows logical top-level guidance that splits the entire analysis process into several distinct analysis stages, to structure the overall reasoning process in a suitable manner. In this study, the stages include initialization, row selection, data type cleaning, reasoning and final answering, while the design of split stages can be freely customized with little effort for the extension in complicated scenarios. At each stage, POTABLE follows a macroscopic instruction to accomplish a precise and specific goal through planning and executing. Consequently, POTABLE reduces the possibility of misleading steps or missing details in the overall reasoning process. In addition, through full code execution under the top-level guidance, it is easy to verify the correctness and completeness of the reasoning process in POTABLE.

## 2.3 Logical Top-level Guidance: Analysis Stage Split

In tabular analysis, a distinguished human analyst follows logical top-level guidance. For instance, they may split the analysis process into several distinct stages in tabular mining and reasoning (Fayyad et al., 1996; Mariscal et al., 2010). Such relatively standard stage splits decompose the overall task goal into more precise and specific sub-goals, allowing more accurate operation planning with less possibility of misleading or missing steps. Inspired by human cognitive behavior in tabular analysis, we integrate logical top-level guidance by splitting the overall procedure into several stages. These stages will be implemented through Python code with `pandas` methods as a common choice of human tabular analysts. Specifically, the overall analysis procedure is split into five stages with macroscopic instruction injection:

- **Initialization**: Store the table data into `pandas.DataFrame` object.
- **Row Selection**: Remove redundant rows that do not represent distinct records.
- **Data Type Cleaning**: Transform the data type of table columns into a suitable form.
- **Reasoning**: Conduct flexible reasoning operations that are useful to find the final answer.
- **Final Answering**: Print out the final answer as the output of the evaluated sample.

In the above stages, the initialization stage is implemented by executing the pre-defined Python code as `import pandas as pd` and `df = pd.DataFrame(data=..., columns=...)`, and then the LLM and the Python interpreter collaborate to traverse the other stages through meticulous planning and execution sequentially. The detailed procedure is explained in the next subsection. Notably, such stage division can be customized easily in different scenarios. We posit that such top-level guidance enhances the reasoning framework to be a distinguished human analyst.

## 2.4 Planning and Executing

To implement the whole table analysis procedure, we adopt a planning phase and an executing phase to complete the macroscopic goal in each stage. Such deployments leverage the LLM's advantage in thinking decomposition and code generation, and enjoy the benefits of robust memorization of structured tables and precise computational results of the symbolic tools simultaneously.

**Planning**. Inspired by Chain-of-Thought (CoT) (Wei et al., 2022) prompting, the LLM decomposes the stage target into operation chains based on the current status of table `df`, while the output is always formatted as `<START>->[OP.]->[OP.]->···-><END>` for easy operation extraction. We do not restrict the scope of planned operations but only require the operations to be useful in achieving the stage target even the overall tabular task goal. To prompt the LLM, we adopt a few-shot learning strategy (Brown et al., 2020) with three self-made examples for the planning phase.

**Executing**. Given an operation, the LLM generates code based on the current table status `df` and the existing code base. For the final answering stage, we adopt few-shot prompting with three self-made examples to obtain the code to print out the answer, while in other stages we adopt zero-shot prompting to generate the code. Next, the generated code is sent to the Python interpreter for real-time execution. Most of the time, the execution is successful and then the table status is updated as the next input from `df` stored in the Python interpreter. Occasionally, the execution fails as the interpreter raises grammar error information or returns illegal output in the final answering stage. In this case, POTABLE will roll back the interpreter to the status before the current execution, and urge the LLM to regenerate suitable code based on the abnormal information.

Consequently, the final answer is obtained from the output of the executed code, instead of the direct LLM response of an LLM query. The overall algorithmic procedure is shown in Algorithm 1.

## 2.5 Summarization

According to the detailed procedure, we can see that POTABLE enjoys two advantages that make it a distinguished human data analyst. **(1) Accuracy**: POTABLE can easily plan coherent operation chains under precise and specific sub-goals at each stage with less possibility of misleading or missing steps through the integration of logical top-level guidance, hence producing more accurate results. **(2) Explainability**: POTABLE follows suitably structured top-level guidance with full

---

**Algorithm 1:** POTABLE

---

**Input:** Table $T$, Question or statement $Q$, LLM $M$ and Python Interpreter $R$
**Output:** Answer $A$ to the question or statement

1  codeBase $\leftarrow$ initalCode($T$)
2  $R$. executeCode(codeBase)
3  **for** stage in { "RowSelection", "DataTypeClean", "Reasoning", "FinalAnswer" } **do**
4      stageModule $\leftarrow$ PoTableBlock(PlanPrompt[stage], CodePrompt[stage])
5      operationList $\leftarrow$ StageModule. plan($T, Q, M$)
6      **for** operaion in operationList **do**
7          code $\leftarrow$ stageModule. codeGen($T, Q, M$, operation, codeBase)
8          errorCnt $\leftarrow$ 0
9          **while** catchError($R$. executeCode(code)) as error and errorCnt $< 10$ **do**
10             $R$. resetEnvironment(). executeCode(codeBase)
11             code $\leftarrow$ stageModule. codeReGen($T, Q, M$, operation, codeBase, error)
12             errorCnt $\leftarrow$ errorCnt $+1$
13         **end**
14         codeBase $\leftarrow$ codeBase $+$ code
15         $T \leftarrow R$. getCurrentStatus($T$)
16     **end**
17 **end**
18 $A \leftarrow R$. getProgramOutput()
19 **return** $A$

---

program execution of each stage and operation, making it easier to verify the completeness and accuracy of the reasoning process along the stage guidance. As a result, POTABLE produces superior reasoning results with high-quality Python programs. The programs are highly accurate, steply commented and completely executable, since they correspond to clear operations and have experienced real-time execution and validation.

## 3 EXPERIMENTS

### 3.1 EXPERIMENTAL SETUP

**Datasets**. We conduct experiments on three evaluation sets of two public benchmarks: **WikiTQ** (Pasupat & Liang, 2015) and **TabFact** (Chen et al., 2020). WikiTQ is a benchmark for table question answering, which requires answering the question with a short corpus based on the given table. We conduct experiments over the validation (dev.) set with 2,831 questions and the test set with 4,344 questions as previous studies do, and use the official denotation accuracy for evaluation. TabFact is a benchmark for table fact verification, which requires judging whether the given statement is true or false based on the given table. We conduct experiments over the released small test set with 2,024 statements as previous studies do, and use the binary classification accuracy for evaluation.

**Backbones**. We select two representative language models as the backbones of POTABLE and other baseline approaches in our experiments. Specifically, we choose GPT-4o-mini (2024-07-18)[1] (**GPT**) as the closed-source small language model, which is competent and cost-efficient to cover a wide range of downstream tasks. In addition, we choose Llama-3.1-70B-Instruct[2] (**LLAMA**) as the open-source LLM for evaluation, which shows strong reasoning capabilities among released foundation models. Please refer to Appendix A for the detailed parameter settings of the backbone models.

**Baselines**. We select four competitive LLM-based approaches as baselines for comparison. **Binder** (Cheng et al., 2023) is a neural-symbolic framework that maps the reasoning task into a specific program and then executes the program binding LLM as a unified API to extend its grammar coverage and tackle the commands that cannot be executed normally. **Dater** (Ye et al., 2023) first decomposes the table into sub-evidence with column and row selection through LLM queries and then decom-

---

[1]https://openai.com/index/gpt-4o-mini-advancing-cost-efficient-intelligence/
[2]https://ai.meta.com/blog/meta-llama-3-1/

Table 1: Accuracy results (%) of table-based reasoning approaches on WikiTQ (**D** denotes dev. set, **T** denotes test set) and TabFact (**S** denotes small test set) on GPT-4o-mini (GPT) and Llama-3.1-70B-Instruct (LLAMA). The best results are marked in **bold** and the second-best results are underlined, while the improvements of POTABLE over the runner-ups are recorded in teal.

| Approach | WikiTQ (D) | | WikiTQ (T) | | TabFact (S) | |
|---|---|---|---|---|---|---|
| | GPT | LLAMA | GPT | LLAMA | GPT | LLAMA |
| Binder (ICLR'23) | 59.20 | 50.65 | 58.86 | 50.51 | 84.63 | 78.16 |
| Dater (SIGIR'23) | 56.76 | 42.78 | 58.33 | 43.53 | 80.98 | 81.57 |
| Chain-of-Table (ICLR'24) | 56.64 | 62.39 | 55.60 | 62.22 | 84.24 | 85.62 |
| TabSQLify (NAACL'24) | 56.87 | 55.51 | 57.02 | 55.78 | 78.75 | 70.70 |
| POTABLE (Ours) | **63.58** | **65.10** | **64.73** | **65.56** | **88.93** | **87.06** |
| | (+4.38) | (+2.71) | (+5.87) | (+3.34) | (+4.30) | (+1.44) |

poses the question into simpler sub-questions through intermediate SQL generation, followed by a joint reasoning stage with simplified tables and questions. **Chain-of-Table** (Wang et al., 2024b) pre-defines several common atomic operations for dynamic selection by the LLM, forming an operation chain to process the table with pre-defined code to simplify the table for the final LLM answer querying. **TabSQLify** (Nahid & Rafiei, 2024) leverages Text-to-SQL to decompose the table into sub-tables and conduct comprehensive reasoning and answer generation through LLM queries. All these selected approaches are competitive as LLM-based baselines on table-based reasoning.

**Implementation Details**. To implement POTABLE, we carefully design prompting templates for planning and executing phases of each stage. In addition, we respectively prepare three few-shot prompting examples for WikiTQ and TabFact, including query-answer pairs for both operation planning and final answer code generation. Please refer to Appendix C for the detailed contents.

## 3.2 MAIN RESULTS

We conduct experiments to compare POTABLE with other baselines over three evaluation datasets of WikiTQ and TabFact on GPT and LLAMA backbones. The result table is presented in Table 1. From the main results, it is clear that our POTABLE significantly outperforms all other baselines over all evaluation datasets from WikiTQ and TabFact on GPT and LLAMA, respectively. In particular, GPT-based POTABLE achieves over 4% higher absolute accuracy than runner-ups on all evaluation datasets, which demonstrates the superior effectiveness of our method.

To be more specific, we make a more comprehensive analysis of results from all approaches and base models. Firstly, Binder is always the runner-up in GPT-based approaches, while its accuracy drop based on LLAMA is 6%-9%. As Binder contains an important step in generating the whole program for the question, it seems that GPT-4o-mini enjoys a higher ability for full code generation. In comparison, POTABLE integrates top-level logical splits of the whole tabular analysis process and generates code once for a single operation with error checking, reducing possible blurred and uncleared code in the overall programs. Consequently, this may be one reason that our method achieves significant improvement over Binder and others in accuracy. Secondly, in LLAMA-based approaches, Chain-of-Table is the second-best approach although it has a constrained operation pool for dynamic selection, while its accuracy drop based on GPT is around 6% in WikiTQ and 1.44% in TabFact. Its reasoning performance mainly depends on the LLM's ability to plan and decompose the operations rather than code generating since the codes for all operations are pre-defined, which may indicate that Llama-3.1-70B-Instruct enjoys a stronger ability to plan and reason. In our POTABLE, the two LLM abilities are fully stimulated, unleashing the potential of symbolic tools for more flexible planning and executing simultaneously. This may be another reason that our method outperforms Chain-of-Table and others in accuracy. Thirdly, in Dater and TabSQLify, the accuracy difference between GPT and LLAMA is unstable across different evaluation datasets. This may indicate that their module and prompt design lack robustness under different LLM bases, while POTABLE demonstrates no such disadvantage.

As a result, the logical top-level guidance integration leads POTABLE to the best accuracy, strongly validating the effectiveness in table-based reasoning scenarios.

Table 2: Accuracy results (%) of POTABLE on different groups in task difficulty as *simple* and *complex* and different groups in table size as *small* (S), *medium* (M) and *large* (L).

| Backbone | Dataset | Task Difficulty | | Table Size | | | Original |
|----------|---------|--------|---------|-------|-------|-------|----------|
| | | Simple | Complex | S | M | L | |
| | WikiTQ (D) | 67.77 | 60.04 | 63.00 | 65.57 | 62.32 | 63.58 |
| GPT | WikiTQ (T) | 68.99 | 61.12 | 70.20 | 66.21 | 61.61 | 64.73 |
| | TabFact (S) | 90.65 | 87.24 | 90.59 | 88.44 | 88.05 | 88.93 |
| | WikiTQ (D) | 68.00 | 62.65 | 68.41 | 67.52 | 61.78 | 65.10 |
| LLAMA | WikiTQ (T) | 68.89 | 62.74 | 71.27 | 67.99 | 61.50 | 65.56 |
| | TabFact (S) | 88.96 | 85.18 | 85.71 | 87.84 | 87.23 | 87.06 |

## 3.3 PERFORMANCE ANALYSIS GROUPED BY TASK DIFFICULTY AND TABLE SIZE

To make further performance analysis of POTABLE, we recompute the main performance at different levels of task difficulty and table size. Specifically, we label the difficulty of the evaluated questions or statements as "simple" or "complex". In WikiTQ, a question with a length less than 50 is labeled as "simple", while a "complex" question is longer. As for TabFact, we use the official difficulty label for all statements. In addition, we group the table content size as "small" (S), "medium" (M) and "large" (L), in situations when the table has 1-49 cells, 50-99 cells and no less than 100 table content cells respectively. The detailed grouped results are reported in Table 2.

The results illustrate that more complex tasks always lead to performance drop as expected, yet the negative correlation between table size and performance is not always obvious. We can draw two preliminary inferences: (1) POTABLE may ignore task decomposition as a potential improvement, although it is trivial to see performance drop on difficult tasks. (2) POTABLE seems somewhat robust on the table size, yet a deeper study grouped by table tokens may be more persuasive.

## 3.4 ABLATION STUDY ON LOGICAL STAGES

In our implementation of POTABLE, the overall tabular analysis procedure is split into five distinct stages. To validate the effect of the logical stage split, we present an ablation study by adopting different stage splits in the compared settings. Specifically, we compare the original GPT-based POTABLE with the following four settings: (1) **Only Reasoning**: we discard all other unnecessary stages except for initialization, reasoning and final answering. In fact, this setting shows no explicit stage split. (2) **Removing Row Selection**: we give up checking redundant rows before further processing and reasoning stages, which is commonly regarded as an operation of sub-table data extraction. (3) **Removing Data Type Cleaning**: we give up checking whether the table column data needs type transformation. As all table columns are stored with an initial type of string, discarding this operation may cause more error execution. (4) **Adding Column Selection**: we add a new column selection stage to select relative columns before further processing and reasoning stages. This stage has been included in most studies as an operation of sub-table data extraction. The overall results of the ablation study on logical stages are shown in Figure 3.

We can see that the original GPT-based POTABLE outperforms all ablated settings in the three evaluation datasets. To be more specific, we make a more comprehensive analysis of results from all settings. Firstly, we focus on the "only reasoning" setting (*only Reason*). Compared with the original setting, *only Reason* scores nearly 0.6%-1% less in WikiTQ and around 3% less in TabFact. These results indicate that logical top-level guidance greatly benefits the tabular analysis process. In addition, *only Reason* shows few weaknesses among all other settings in WikiTQ but has more accuracy drop in TabFact. From the task perspective, WikiTQ is about a clear task to answer the question directly, while TabFact asks to judge the statement, containing intermediate reasoning processes to judge the potential sub-facts. Therefore, the logical split for TabFact may be more necessary than WikiTQ, and it is also crucial to make the split as reasonable as possible.

On the other hand, the results of settings "removing row selection" and "removing data type cleaning" demonstrate the importance of these stages as expected. Notably, "removing data type cleaning" always reaches the lowest accuracy, indicating that the framework may ignore these details (*e.g.*, treating text floats into real floats) with unreasonable top-level stage-split. A more interesting

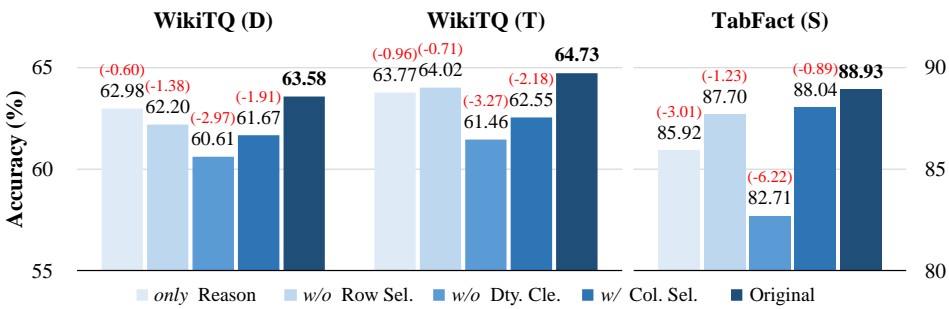

Figure 3: Accuracy results (%) in the ablation study of the different logic split employed in POTABLE with GPT-4o-mini on three evaluation sets of WikiTQ and TabFact, including only reasoning (*only* Reason), removing row selection (*w/o* Row Sel.), removing data type cleaning (*w/o* Dty. Cle.), adding column selection (*w/* Col. Sel.) and the original setting (Original). The best results are marked in **bold**, while the accuracy drops in all settings are recorded in red.

Table 3: Efficiency results on TabFact (S) for GPT-based methods. For the three baselines, we compared the results of single LLM generation (*Single*) and default LLM generation (*Default*) following their claimed settings in the article. Here *Gen.* denotes "generation" and *ave.* denotes "average".

| Approach | Accuracy | | # Generation (Default) | Details |
|---|---|---|---|---|
| | Single | Default | | |
| Binder | 84.63 | 85.13 | 50 | SQL *Gen.*: 50 |
| Dater | 80.98 | 82.26 | 100 | Decomposition *Gen.*: 40, Cloze *Gen.*: 20, SQL *Gen.*: 20, Query: 20 |
| Chain-of-Table | 84.24 | 85.23 | ≤22 | Dynamic Planning: ≤4 (3.74 on *ave.*), Args *Gen.*: ≤17 (16.09 on *ave.*), Query: 1 |
| POTABLE (*only* Reason) | 85.92 | | ≤6 | Planning: 1, Code *Gen.*: ≤ 4 (3.72 on *ave.*), Re-*Gen.*: ≤1 (less than 1 on average) |
| POTABLE | **88.93** | | ≤10 | Planning: 3, Code *Gen.*: ≤ 6 (5.60 on *ave.*), Re-*Gen.*: ≤1 (less than 1 on *ave.*) |

fact is the addition of "column selection" results in worse performance, which is widely adopted in previous approaches. We speculate that the selected backbones are competitive enough to handle full table columns, yet eliminating seemingly irrelevant columns may cause a dilemma, *i.e.*, potentially useful columns are accidentally removed and the LLM reasoner cannot find adequate data for processing. Therefore, column selection is not suitable to be regarded as a distinct stage.

As a completion, we also recompute the performance results grouped by task difficulty and table size. Please refer to Appendix B to check the detailed results and analyses.

## 3.5 EFFICIENCY ANALYSIS

We analyze the efficiency of POTABLE and three representative baselines based on GPT by evaluating the count number of required LLM-based generation in TabFact (S). The result tables are presented in Table 3. We notice that the multiple generation achieves some improvement in the compared baselines, yet PoTable always adopts the single generation and outperforms them. In Binder and Dater, the generation counts are fixed while the ones of Chain-of-Table and our PoTable fluctuate dynamically. Therefore, we report the empirical average counts of each module and rounded them up as their estimation. It can be seen that POTABLE has much fewer LLM generation counts than previous baselines. In addition, the difference in generation counts between the original POTABLE and the *only Reason* setting is small, while the accuracy improvement is more than 3%. These results demonstrate the efficiency of our POTABLE, indicating that the improvements come from the top-level guidance integration rather than multiple generations.

| rank | country | box office | year | box office from national films |
|------|---------|------------|------|-------------------------------|
| 1 | Canada/United States | $10.8 billion | 2012 | \u2013 |
| 2 | China | $3.6 billion | 2013 | 59.7% (2013) |
| ... | ... | ... | ... | ... |
| 5 | France | $1.7 billion | 2012 | 33.3% (2013) |
| 6 | South Korea | $1.47 billion | 2013 | 59.7% (2013) |
| ... | ... | ... | ... | ... |
| 12 | Brazil | $0.72 billion | 2013 | 17% (2013) |
| - | World | $34.7 billion | 2012 | \u2013 |

**Q:** Who ranks after France in the list of largest markets in the film industry by box office?

**A:** South Korea    **Program Output:** South Korea ✔

**Initialization (Pre-defined)**
```python
import pandas as pd
df = pd.DataFrame(data=[...], columns=[...])
```

**Row Selection**
```python
# remove rows where `rank`='-'
df = df[df['rank'] !='-']
```

**Data Type Cleaning**
```python
# transform column `rank` into `int` type
df['rank'] = df['rank'].astype(int)
```

**Reasoning**
```python
# extract the rank of france
france_rank = df.loc[df['country'] ==
'france', 'rank'].values[0]

# find the country that has a rank one
greater than the rank of france
next_rank = france_rank + 1
next_country = df.loc[df['rank'] ==
next_rank, 'country'].values[0]
```

**Final Answering**
```python
# final output
print(next_country)
```

Figure 4: A case study of an evaluated sample from WikiTQ (T) wit its generated Python program and output answer, which indicates the effectiveness and explainability of POTABLE.

## 3.6 CASE STUDY

We conduct a case study of POTABLE in Figure 4 by presenting an evaluated sample from WikiTQ (T) with its generated Python program and output answer. The tabular task sample is fed into POTABLE, experiencing a relatively standard analysis process including five logical stages. From the complete program, we notice the planned operations (shown as split comments in the stage block) and high-quality generated code of each operation (matching the former comment) for real-time execution. POTABLE follows suitably structured top-level guidance with full program execution of each stage and operation, allowing us to easily review the whole process precisely and discover the true reason why it leads to right or wrong answers. Along with the answer, POTABLE produces highly accurate, steply commented and completely executable code. These produced outputs demonstrate that POTABLE enjoys high accuracy and explainability.

## 4 RELATED WORK

**Table Processing with Language Models**. Table processing has been a popular research domain over the past decade. Before the era of LLMs, numerous efforts were made to process tables with pre-trained language models. TaPas (Herzig et al., 2020) extends BERT (Devlin et al., 2019) by conducting masked pre-training with joint encoding of questions and flattening tables. TaBERT (Yin et al., 2020) combines content snapshot and vertical attention based on BERT to obtain joint textual and tabular representations for further understanding. TUTA (Wang et al., 2021) enhances transformers (Vaswani et al., 2017) with structure-aware mechanisms to effectively capture spatial, hierarchical and semantic information. TAPEX (Liu et al., 2022) pre-trains BART (Lewis et al., 2020) on a large synthetic SQL dataset to imitate the SQL executor that better understands tabular structure information. With the development of LLMs, the paradigm of table processing has been deeply revolutionized, especially in tabular data encoding and reasoning. In prompting methods, Sui et al. (2024) designs a benchmark to evaluate the structural understanding capabilities of LLMs, followed by a novel self-augmentation for effective structural prompting. Dater (Ye et al., 2023) and DIN-SQL (Pourreza & Rafiei, 2023) adopt task decomposition for better understanding with simplified queries, while Chain-of-Table (Wang et al., 2024b) defines atomic operations for dynamic selection in CoT prompting. Some other approaches explore training or tuning LLMs as generalists. TableLlama (Zhang et al., 2024a) develops an open-source tabular LLM by fine-tuning Llama 2-7B (Touvron et al., 2023) with LongLoRA (Chen et al., 2024b), while Table-LLAVA (Zheng et al., 2024) trains a multi-modal tabular LLM that can handle table images as vision inputs.

**Table Processing with Symbolic Tools**. Symbolic tools have been widely utilized as assistants to produce more accurate and robust mid-results in LLM-based table reasoning scenarios. Most studies adopt databases and Python as affiliated executors to interact with LLMs. Binder (Cheng et al., 2023) and Cao et al. (2023) parse the tasks into integral SQL or Python programs for further ex-

ecution, incorporating LLM-assistant APIs to handle abstract code blocks for complete execution. TabSQLify (Nahid & Rafiei, 2024) generates SQL queries to extract sub-tables and executes them to get simplified tables for further LLM reasoning. Some works target boosting the code generation ability for tabular reasoning and other scenarios. TroVE (Wang et al., 2024a) asks the code LLMs to curate reusable high-level functions and use them to write solutions for Python execution on the table question answering and other tasks, while Self-Debugging (Chen et al., 2024a) teaches LLMs to debug their predicted SQL or Python programs on Text-to-SQL (Yu et al., 2018) and other tasks. Recently, research in LLM-based table reasoning has been extended into more sophisticated tools environments and more advanced reasoning tasks. SheetCopilot (Li et al., 2023a) and Spreadsheet-Bench (Ma et al., 2024) address a novel spreadsheet manipulation task, which maneuvers table analysis software like Microsoft Excel[3] to generate step-by-step solutions for simulated execution. MatPlotAgent (Yang et al., 2024) addresses the task of scientific data visualization, which includes a code agent integrating `Matplotlib`[4] responsible for generating the code to plot figures from input tables. As a result, symbolic tool utilization has become a crucial component in table processing.

## 5 CONCLUSION

In this paper, we proposed POTABLE as a simple yet effective table-based reasoning method. POTABLE featured a planning phase and an executing phase implemented by an LLM and a Python interpreter, incorporating logical top-level guidance through analysis stage splitting with macroscopic instruction injection. Consequently, POTABLE produced highly accurate, steply commented and completely executable code to obtain reliable answers. Accordingly, POTABLE enjoyed two advantages of high accuracy and explainability, making it a distinguished tabular data analyst. Extensive experiments under three evaluation datasets of two benchmarks on different backbones presented a dominating performance of POTABLE on table-based reasoning.

This study targeted the balance of structure and autonomy through suitable top-level guidance integration in standardized table-based reasoning. However, more complicated tabular data (*e.g.*, hierarchical tables, multiple tables) and more domain-specific scenarios (*e.g.*, spreadsheet manipulation, healthcare records) remained less explored. In the future, we will explore more effective ways to make our improved method competent on more complicated and domain-specific table-based reasoning scenarios, simulating more advanced human behavior in tabular analysis.

## REPRODUCIBILITY STATEMENT

Our code is available in `https://anonymous.4open.science/r/PoTable-6788` for reproducibility. All baseline approaches have released the official open-source code and prompts. Specifically, we run Binder from `https://github.com/xlang-ai/Binder`, Dater from `https://github.com/AlibabaResearch/DAMO-ConvAI/tree/main/dater`, Chain-of-Table from `https://github.com/google-research/chain-of-table`, and TabSQLify from `https://github.com/mahadi-nahid/TabSQLify`.

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

# APPENDIX

## A PARAMETER SETTINGS OF BACKBONES

We report the parameter settings of GPT-4o-mini (2024-07-18) and Llama-3.1-70B-Instruct as the backbone models for POTABLE in Table 4. In all logical stages of the three evaluation datasets in WikiTQ and Tabfact, the parameter setting remains unchanged. As for the baselines, we only unify the setting of n_samples as 1 for a fair generation effect comparison, while for other parameters, we use the originally proposed settings since the targeted operations and the deployed paradigm in different approaches are different.

Table 4: Parameter settings of GPT and LLAMA backbone models in POTABLE.

| Backbone | temperature | top_p | max_tokens | n_samples |
|---|---|---|---|---|
| GPT | 0.1 | 0.9 | 2,048 | 1 |
| LLAMA | 0.1 | 0.9 | 2,048 | 1 |

## B FINE-GRAINED ABLATION STUDY RESULTS

Following the same group division in comparison experiments, we report the fine-grained results of the ablation study in GPT-based POTABLE. The result table grouped by task difficulty is shown in Table 5, while the one grouped by table sizes is shown in Table 6. In most of the time, the original setting reaches the best results in different grouped settings, strengthening the conclusions from the ablation study in the main text. In addition, these results strongly indicate the influence of task difficulty but do not seem to be that strong on table size on cells, as illustrated in the main text.

Table 5: Fine-grained accuracy results (%) in the ablation study grouped by different task difficulty as *simple* and *complex* in GPT-based POTABLE.

| Setting | WikiTQ (D) | | WikiTQ (T) | | TabFact (S) | |
|---|---|---|---|---|---|---|
| | Simple | Complex | Simple | Complex | Simple | Complex |
| only Reason | 66.38 | **60.10** | 66.98 | 60.19 | 85.87 | 85.97 |
| w/o Row Sel. | 66.23 | 58.80 | 67.74 | 60.87 | 88.66 | 86.75 |
| w/o Dty. Cle. | 63.99 | 57.76 | 65.48 | 58.06 | 80.70 | 84.69 |
| w/ Col. Sel. | 65.30 | 58.60 | 66.18 | 59.46 | 88.96 | 87.14 |
| **Original** | **67.77** | 60.04 | **68.99** | **61.12** | **90.65** | **87.24** |

Table 6: Fine-grained accuracy results (%) in the ablation study grouped by different table sizes as *small* (S), *medium* (M) and *large* (L) in GPT-based POTABLE.

| Setting | WikiTQ (D) | | | WikiTQ (T) | | | TabFact (S) | | |
|---|---|---|---|---|---|---|---|---|---|
| | S | M | L | S | M | L | S | M | L |
| only Reason | **65.45** | 64.14 | 61.00 | 68.82 | 64.59 | 61.35 | 85.89 | 86.53 | 85.11 |
| w/o Row Sel. | 61.95 | 64.24 | 60.76 | 69.74 | 65.28 | 60.99 | 87.80 | 87.37 | **88.05** |
| w/o Dty. Cle. | 60.73 | 62.70 | 58.97 | 66.67 | 63.50 | 57.92 | 83.80 | 83.55 | 80.52 |
| w/ Col. Sel. | 58.99 | 64.24 | 60.92 | 67.28 | 65.05 | 58.74 | 89.02 | 87.49 | 87.89 |
| **Original** | 63.00 | **65.57** | **62.32** | **70.20** | **66.21** | **61.61** | **90.59** | **88.44** | **88.05** |

## C  IMPLEMENTATION DETAILS OF POTABLE

We list all prompt templates in Figure 5-22. These prompt templates are combined based on different stages and scenarios in the planning and executing modules.

In operation planning, the prompt templates are combined as follows:

- **Row Selection Stage**: Figure 7/8 (WikiTQ/TabFact) + Figure 9.
- **Data Type Cleaning Stage**: Figure 5/6 (WikiTQ/TabFact) + Figure 10.
- **Reasoning Stage**: Figure 5/6 (WikiTQ/TabFact) + Figure 11/12 (WikiTQ/TabFact).
- **Column Selection Stage (Ablation)**: Figure 5/6 (WikiTQ/TabFact) + Figure 13.

In code generation for execution, the prompt templates are combined as follows:

- **Row Selection Stage**: Figure 7/8 (WikiTQ/TabFact) + Figure 14.
- **Data Type Cleaning Stage**: Figure 5/6 (WikiTQ/TabFact) + Figure 15.
- **Reasoning Stage**: Figure 5/6 (WikiTQ/TabFact) + Figure 16/17 (WikiTQ/TabFact).
- **Column Selection Stage (Ablation)**: Figure 5/6 (WikiTQ/TabFact) + Figure 14.
- **Non-final Code Regeneration**: Figure 18.
- **Final Answering Stage**: Figure 5/6 (WikiTQ/TabFact) + Figure 19/20 (WikiTQ/TabFact).
- **Final Code Regeneration**: Figure 21/22 (WikiTQ/TabFact).

In addition, we have constructed three samples based on the table of Paris 2024 Olympic Medal Count, which are utilized for few-shot prompting to generate the planning operations list and the final answering code. As for code generation and regeneration in other situations, we adopt zero-shot prompting. To check the specific contents of the demo samples, please refer to our code repository.

```
Given the table information:
/*
Data:
{table_df}
*/
Here is a statement to be answered:
/*
Statement: {question}
*/
```

Figure 5: The prompt template of table information in WikiTQ.

```
Given the table information:
/*
Caption: {caption}
Data:
{table_df}
*/
Here is a statement to be verified:
/*
Statement: {statement}
*/
```

Figure 6: The prompt template of table information in TabFact.

```
Given the table information:
/*
Data:
{table_df}
*/
```

Figure 7: The prompt template of table information without the question in WikiTQ.

```
Given the table information:
/*
Caption: {caption}
Data:
{table_df}
*/
```

Figure 8: The prompt template of table information without the statement in TabFact.

```
INSTRUCTION:
Judge if there are redundant rows that can be obtained from other
independent row data. For example, if the statement do not mention words
like `total`, `average`, rows like `total`, `average` that do not
represent a distinct item data, should be removed.

FORMAT:
"<START> -> [OPERATION] -> <END>"

NOTE:
1. If no such rows exist, skip this [OPERATION] and generate "<END>"
directly to finish the plan. This operation should be generated in most
of the time even if you are not certain.
2. If such rows exist, remove them. In this case the format of this
[OPERATION] should be "remove rows where `XXX`=`YYY`, ...", here `XXX` is
the name of the first column, and `YYY` is the corresponding value (e.g.,
`total`, `average`). This [OPERATION] should be generated only once when
you are very confident that the rows are redundant.

OUTPUT:
{output}
```

Figure 9: The planning prompt template of row selection stage.

```
INSTRUCTION:
All columns of the table stored in pandas.DataFrame `df` are string type.
Judge if there exist columns that need data type transformation. If so,
generate a plan to transfer the corresponding column type.

FORMAT:
"<START> -> [OPERATION] -> ... -> [OPERATION] -> <END>"

NOTE:
1. You can transfer the columns with integer values into `int` data type
or the columns with real number values into `double` type. The format of
this [OPERATION] should be "transfer column `XXX` into `XXX` type".
2. If there is no need to perform, skip this [OPERATION] and generate
"<END>" directly to finish the plan.

OUTPUT:
{output}
```

Figure 10: The planning prompt template of data type cleaning stage.

```
INSTRUCTION:
Generate a reasoning plan that can be easily executed by python code, to
answer the given statement.

FORMAT:
"<START> -> [OPERATION] -> ... -> [OPERATION] -> <END>"

NOTE:
Candidate [OPERATION] contain column value sorting, conditional data
counting, arithmetic calculations, expression comparison and other
reasoning operations, etc.

OUTPUT: {output}
```

Figure 11: The planning prompt template of reasoning stage in WikiTQ.

```
INSTRUCTION:
Generate a reasoning plan that can be easily executed by python code, to
verify whether the statement is true.

FORMAT:
"<START> -> [OPERATION] -> ... -> [OPERATION] -> <END>"

NOTE:
Candidate [OPERATION] contain column value sorting, conditional data
counting, arithmetic calculations, expression comparison and other
reasoning operations, etc.

OUTPUT: {output}
```

Figure 12: The planning prompt template of reasoning stage in TabFact.

```
INSTRUCTION:
Select columns that are somewhat relevant in semantics to the statement.

FORMAT:
"<START> -> [OPERATION] -> <END>"

NOTE:
1. The first column should be always selected.
2. The format of this [OPERATION] should be "select columns named
`XXX`, ...".
3. This [OPERATION] should be generated only once.

OUTPUT:
{output}
```

Figure 13: The planning prompt template of column selection stage.

```
We have executed the following code:
```python
{code_base}
```
Now we need to continue to execute the following operation: {operation}

INSTRUCTION:
Generate code without any other texts according to the given operation.

FORMAT:
```python
df = XXXXXX
```

NOTE: The table is stored in a pandas.Dataframe variable named `df`.

OUTPUT: {output}
```

Figure 14: The code generation prompt template of row selection and column selection stage.

```
We have executed the following code:
```python
{code_base}
```
Now we need to continue to execute the following operation: {operation}

INSTRUCTION:
Generate code without any other texts according to the given operation.

FORMAT:
```python
df['XXX'] = XXXXXX
```

OUTPUT: {output}
```

Figure 15: The code generation prompt template of data type cleaning stage.

```
We have executed the following code:
```python
{code_base}
```
Now we need to continue to execute the following operation: {operation}

INSTRUCTION:
Generate code without any other texts according to the given operation.
Remember to store the result into suitable variables.

FORMAT:
```python
[GENERATED CODE]
```

NOTE:
Do not store any formatted strings. For example, if the answer of winner
is "John", then just store "John" directly instead of formatted strings
like "John is the winner", "John wins". In addition, if the answer of
country number is "0", then just store "0" directly instead of formatted
strings like "no countries", "there is no countries".

OUTPUT: {output}
```

Figure 16: The code generation prompt template of reasoning stage in WikiTQ.

```
We have executed the following code:
```python
{code_base}
```
Now we need to continue to execute the following operation: {operation}

INSTRUCTION:
Generate code without any other texts according to the given operation.
Remember to store the result into suitable variables.

FORMAT:
```python
[GENERATED CODE]
```

OUTPUT: {output}
```

Figure 17: The code generation prompt template of reasoning stage in TabFact.

```
When executing the generated code, the python interpreter raises the
following error information:
{output}
INSTRUCTION: Please regenerate legal code for the given operation.
```

Figure 18: The code generation prompt template of regeneration.

```
We have executed the following code:
```python
{code_base}
```

INSTRUCTION:
Based on the executed code, continue to generate the final output code to
print out the variable indicating the answer of the statement. The
variable should be one of `int`, `float`, `string`, `bool` type or a list
containing elements of these types. Remember to use `print()` method in
the generated code.

FORMAT:
```python
...
print(XXX)
```

NOTE:
Here `XXX` denotes the variable indicating the answer of the statement.
Do not print out any irrelavent variables or strings.

OUTPUT: {output}
```

Figure 19: The code generation prompt template of final answering stage in WikiTQ.

```
We have executed the following code:
```python
{code_base}
```

INSTRUCTION:
Based on the executed code, continue to generate the final output code to
print out the bool type variable indicating whether the statement is true
or not. Remember to use `print()` method in the generated code.

FORMAT:
```python
...
print(XXX)
```

NOTE:
Here `XXX` denotes the bool type variable or boolean expression
indicating whether the statement is true or not.

OUTPUT: {output}
```

Figure 20: The code generation prompt template of final answering stage in TabFact.

```
When executing the generated code, the python interpreter has the
following output:
{program_output}
It is an illegal type variable or a blank string/list, which is not
acceptable.

INSTRUCTION: Please regenerate legal code to print out the corresponding
variable indicating the answer of the statement. The variable should be
one of `int`, `float`, `string`, `bool` type or a list containing
elements of these types. Remember to use `print()` method in the
generated code.

FORMAT:
```python
...
print(XXX)
```

NOTE:
Here `XXX` denotes the variable indicating the answer of the statement.

OUTPUT:
```

Figure 21: The code generation prompt template of final answering regeneration in WikiTQ.

```
When executing the generated code, the python interpreter has the
following output:
{program_output}
It is neither True or False that indicates whether the statement is true
or not.

INSTRUCTION:
Please regenerate legal code to print out the bool type variable
indicating whether the statement is true or not. Remember to use
`print()` method in the generated code.

FORMAT:
```python
...
print(XXX)
```

NOTE:
Here `XXX` denotes the bool type variable or boolean expression
indicating whether the statement is true or not.

OUTPUT:
```

Figure 22: The code generation prompt template of final answering regeneration in TabFact.

