# OpenReview forum: "PoTable: Programming Standardly on Table-based Reasoning Like a Human Analyst"
_ICLR.cc/2025/Conference — Submitted to ICLR 2025_

### Official Review · Reviewer_wUJL · 2024-10-23

**Soundness:** 3
**Presentation:** 4
**Contribution:** 3
**Rating:** 6
**Confidence:** 4

**Summary:**

The study introduces PoTable, a novel method for table-based reasoning that effectively simulates human cognitive behavior by integrating a Python interpreter with a Large Language Model (LLM). This approach addresses limitations in previous methods by employing a human-like logical stage split and expanding the operation pool into an open-world space. PoTable uses an LLM-based operation planner and code generator to produce highly accurate, step-by-step commented, and fully executable programs. The method demonstrates strong effectiveness and explainability, achieving over 4% higher accuracy than competitors across multiple datasets.

**Strengths:**

- Tabular analysis is crucial for both research and practical applications, highlighting the relevance of this study.
- The proposed method, POTABLE, achieves state-of-the-art performance on selected evaluation datasets, outperforming existing approaches.

**Weaknesses:**

1. I don't think Initialization and Final Answering are important enough to be count as stages.
2. The claim that your analysis stage split is human-like is too arbitrary. How do you know that human do tabular analysis in this way? Have you ever done any research about this? if so, please cite them in the paper.
3.  The evaluation datasets are not extensive enough. Evaluation on just two datasets cannot really support your concusion. For evaluation datasets, why did you choose to use both validation and test sets of WiKiTQ? is there any inherent difference between these two sets? The evaluation results in Table 1 actually show almost similar results for both sets.
4. The results analysis in the seconde paragraph in Sec 3.2 could be further improved. I think it is important to discuss the phenomenon that the difference between GPT and LLAMA is quite different across approaches, i.e., for WiKiTQ, GPT > LLAMA for Binder, Dater, GPT < LLAMA for Chain-of-Table, GPT = LLAMA for TabSQLify and PoTable. The same for TabFact.
5. In Figure 3, it is suggested to discuss why "removing data type cleaning (w/o Dty. Cle.)" gets the lowest scores. It would be weird if you ignore this result because it is so obvious in the figure.
6. I found that the max tokens of the backbone models used in the study is 2048 (Appendix A). Thus i am particular interested in that, if you put the full table in the prompt, how do handle the table with large content exceeding 2048 tokens?  Also, the code_base would also increase as the analysis stage goes ahead.
7. In figure 11, the prompt contains several suggested operations when generating the reasoning plan. Would this be contradicted with the claim that the PoTable "extends the operation pool into an open-world space without any constraints."? If not, please provide some examples to prove.

**Questions:**

See questions in Weaknesses.

---

> ### Author Response · Authors · 2024-11-23
> **Official Response to Reviewer wUJL [1/2]**
>
> Thank you for your constructive feedback. We have carefully considered your suggestions and revised our manuscript. Here are our detailed responses to your concerns.
>
> > **W1.** I don't think Initialization and Final Answering are important enough to be count as stages.
>
> We understand why you have raised this concern.
> In our evaluated tasks, the initialization stage used pre-defined code to store a standardized single table, while the final answering stage only needed to output the final reasoning answer.
> In fact, in some complex scenarios, these two stages require some more refined adjustments.
> For example, when storing hierarchical or multiple tables, we may require more complex planning and code generation in the initialization stage.
> The processing of multi-hop question answering and multiple key variable combinations follows the same principle.
> In order to further expand in more complex scenarios in the future, we define these two parts as distinct stages.
>
> > **W2.** The claim that your analysis stage split is human-like is too arbitrary. If is actually does, please cite evidence in the paper.
>
> We would apologize for our previous expression that have impacted your understanding of our study.
> This prompted us to rethink and refine the our motivation and contributions.
> Specifically, we overemphasized that our stage split was **standard** rather than conveying more essential concept of **integrating logical top-level guidance**.
> In this way, PoTable enjoys two advantages that make it a distinguished data analyst. **(1) Accuracy**: PoTable can easily plan coherent operation chains under precise and specific sub-goals with less possibility of
> misleading or missing steps, producing more accurate results. **(2) Explainability**: PoTable follows suitable structured top-level guidance with full operation execution, making it easier to verify the completeness and accuracy of the reasoning process.
> Furthermore, the logical stage split was inspired by human data analysis and mining processes [1][2], aiming to make the reasoning results as accurate and explainable as possible.
> In the revised version, we will describe the stage split as "relatively standard" because it indeed represents a widely-accepted top-level guidance among most researchers.
> We sincerely thank you for your suggestions, which have significantly improved the motivation and contributions of our work.
>
> > **W3.** The evaluation datasets are not extensive enough.
>
> Some previous studies (*e.g.*, Dater, Chain-of-Thought) evaluate their method through three commonly-used datasets, *i.e.*, WikiTQ, TabFact and FeTaQA since they all adopt LLM-query as the main reasoning mode.
> However, PoTable adopts full program execution to generate answers, being not able to handle free-form answer generation in FeTaQA directly.
> Therefore, we only adopt WikiTQ and TabFact as the evaluation datasets.
> Following Dater, we also evaluate the dev set of WikiTQ, and the fine-grained ablation study results in fact reveal that the dev set follow somewhat shifted distribution towards the test set.
>
> > **W4.** The results analysis in the seconde paragraph in Sec 3.2 could be further improved.
>
> Thank you for your suggestion.
> The detailed analysis of Binder and Chain-of-Table has already presented in the current manuscript.
> As for Dater and TabSQLify, we indeed observe there unstable accuracy difference across different LLM bases and datasets.
> Based on their implementation code and results, we can only make assumptions that their module and prompt design lack robustness under different LLM bases, while PoTable demonstrates no such disadvantage.
> We will revise the corresponding analysis in the new revised manuscript.
>
> > [1] Fayyad U, Piatetsky-Shapiro G, Smyth P. The KDD process for extracting useful knowledge from volumes of data[J]. Communications of the ACM, 1996, 39(11): 27-34.
>
> > [2] Mariscal G, Marban O, Fernandez C. A survey of data mining and knowledge discovery process models and methodologies[J]. The Knowledge Engineering Review, 2010, 25(2): 137-166.

---

> ### Author Response · Authors · 2024-11-23
> **Official Response to Reviewer wUJL [2/2]**
>
> > **W5.** It is suggested to discuss why "removing data type cleaning (w/o Dty. Cle.)" gets the lowest scores.
>
> Thank you for your suggestion.
> Among all ablated settings, *w/o* Dty. Cle. always reaches the lowest accuracy.
> This is because, in our predefined code, all cell contents are stored as string types.
> In a no-stage-split setting (only Reason), the code generation module can easily consider the need for data type cleaning for this part given the initial code base of table variable definition.
> However, in *w/o* Dty. Cle. setting, such cleaning can only be performed after the row selection stage, which leads to more content in the code base so that the planner and code generator may find it easier to ignore such details of type cleaning.
> Therefore, the results indicate that data type cleaning is a very important stage, and the framework may ignore details with unreasonable top-level stage split.
> We will revise the corresponding analysis in the new revised manuscript.
>
>
>
> > **W6.** The max tokens of the backbone models used in the study is 2048, with possible risks of exceed input.
>
> In fact, the `max_tokens` restricts the **output** token number of LLM generations instead of the input one.
> Our selected base LLMs accept 128K input tokens, so there is no risk of exceeding input.
>
> > **W7.** Would suggested operations be contradicted by the claim of "open-world space without any constraints"?
>
> In fact, the suggested operations in prompts were not contradicted by the claim of "open-world space without any constraints".
> During the reasoning stage, PoTable may plan and execute some operations that are not recommended, such as extracting some intermediate variables, as shown in the case study.
> Although we will no longer use the expression "open-world space" in the revised manuscript, we still want to point out that the relevant hints are only to make the planner aware of what should be done in the reasoning stage.
> In this way, PoTable integrates logical top-level guidance with adequate autonomy in tool-based execution.

---

> > ### Comment · Reviewer_wUJL · 2024-11-26
> > **Follow-up comments for authors**
> >
> > Thank you for your response. Here are my follow-up comments:
> > > Response to W1
> >
> > This response provides some support for the importance of Initialization stage. However, support for Final Answering stage is still missing.
> > > Response to W2
> >
> > This response is fine, though there is too much irrelevant content.
> > > Response to W3
> >
> > I am not familiar with the table analysis area, so two test sets *may be enough* in this area as claimed by authors. However, I still did not see any necessary to test both dev and test sets of WikiTQ. I did not see any "the dev set follow somewhat shifted distribution towards the test set." in the ablation study.
> > > Response to W4
> >
> > This response addressed my question.
> > > Response to W5
> >
> > This response addressed my question.
> > > Response to W6
> >
> > This response addressed my question. I still want to point out that context window is always a factor you need to consider when analyzing tables, especially if it is a huge table when even 128k cannot handle. How will you deal with this situation?
> > > Response to W7
> >
> > If you agree to not use the expression "open-world space" in the revised manuscript, then please also remove related contents in the abstract and contributions. That means, "open-world space without any constraints" should not be emphasized as a highlight of your work.
> >
> > I am willing to discuss further with authors if you seriously consider my comments. I will consider increasing the rating score if the follow-up responses are satisfactory.

---

> > > ### Author Response · Authors · 2024-11-27
> > > **Follow-up Responses to Reviewer wUJL**
> > >
> > > Thanks for your follow-up reply. Here are our additional explanations:
> > >
> > > > **W1**. Support for *Final Answering* stage is still missing.
> > >
> > > In fact, the initial version of PoTable just combines the current *reasoning* stage and *final answering* stage, yet we have observed that it cannot handle questions with multiple subfacts well.
> > > For instance, given the statement "John of France never won Jack of Italy", the reasoning part may store three variables as `v1 = John comes from France`, `v2 = # John won Jack` and `v3 = Jack from Italy`.
> > > However, the initial PoTable may:
> > > - Just prints `v3` out that ignores other subfacts.
> > > - Prints `v1 and v2` out that forgets to verify if `v2 == 0`.
> > > - Only generates code without `print()`, so we get nothing from the executed program.
> > >
> > > The possible reason is that in the *reasoning* stage, PoTable generates **too many operations that may lose some crucial details**, and this **output goal will be blurred** for the executor.
> > > Therefore, we split *final answering* as a distinct stage with clear instructions that expect PoTable to print out correct and reasonable answers (*e.g.*, print `v1 and v2 == 0 and v3` in the above instance).
> > > Since the planner will generate a few operations under a precise and clear stage goal, it is less likely to make mistakes. Empirical experiments also demonstrate it.
> > >
> > > We hope to well improve the stage as the role of joint summarizer or concluder in more complex scenarios (*e.g.*, long-form TQA that requires answering with long texts) in future studies.
> > >
> > > > **W3**. Necessary to test both the dev and test sets of WikiTQ.
> > >
> > > According to the official paper of WikiTQ [1], this dataset addresses the increased breadth in the knowledge source to generate logical forms from **novel tables with previously unseen relations and entities**.
> > > In its official repository (especially in [README](https://github.com/ppasupat/WikiTableQuestions/blob/master/README.md#questions-and-answers)), the test data set refers to `pristine-unseen-tables`, whose tables **are not seen in training data**.
> > > As for the dev set, it refers to `random-split-1-dev` which is the 20\% dev part of an 80-20 split in `training.tsv` (Dater [2] selected this specific split dev set for evaluation), hence they follow a similar distribution to the training set.
> > > Therefore, the dev set and the test set follow a different distribution without any overlapped tables, and you can regard it as an augmented evaluation.
> > >
> > > In addition, the fine-grained ablation results in Table 5 and 6 in the Appendix (page 14) show that the performance ordering of the compared settings are different in the dev set and test set, indicating some possible distributional differences.
> > >
> > > > **W6**. The case when the tables are larger than 128K.
> > >
> > > This is quite a good suggestion.
> > > In fact, as the main tables are stored in the `pandas.DataFrame` variables, there is a simple and effective approach to just encode **the first several rows of the tables** into the agent so that it can understand the data format of every table column through few-shot examples, to make plans with possibly little performance drop.
> > > Since the evaluated datasets in this study show no such case, the current PoTable does not employ this improvement, for a full evaluation performance.
> > > We will also consider this point to handle large tables in our future study.
> > >
> > > > **W7**. Elimination of "open-world space" in the revised manuscript.
> > >
> > > We strongly agree with your suggestion. We have already submitted a revised manuscript to the system and removed all contents related to *open-world space*, which will no longer be a highlight point.
> > >
> > > We hope the additional explanations will better dispel your concern.
> > >
> > > > [1] Pasupat P, Liang P. Compositional Semantic Parsing on Semi-Structured Tables[C]. Proceedings of the 53rd Annual Meeting of the Association for Computational Linguistics and the 7th International Joint Conference on Natural Language Processing (Volume 1: Long Papers). 2015: 1470-1480.
> > >
> > > > [2] Ye Y, Hui B, Yang M, et al. Large language models are versatile decomposers: Decomposing evidence and questions for table-based reasoning[C]. Proceedings of the 46th International ACM SIGIR Conference on Research and Development in Information Retrieval. 2023: 174-184.

---

> > > > ### Comment · Reviewer_wUJL · 2024-11-27
> > > > **I increased my rating**
> > > >
> > > > I appreciated your responses. The responses have addressed most of my questions. Thus I have raised my rating.
> > > > However, I still am concerned the use of the dev and test sets of WikiTQ. Just a kind remainder that it would be better to find an total independent test set as the third one. Thanks.

---

### Official Review · Reviewer_Xpqj · 2024-10-27

**Soundness:** 2
**Presentation:** 3
**Contribution:** 2
**Rating:** 3
**Confidence:** 4

**Summary:**

This paper proposes a method, PoTable, to solving table-based reasoning tasks with two contributions: (i) logically standard reasoning and (ii) unrestricted operation tools.

**Strengths:**

1. The proposed method appears to be effective on the three datasets experimented.

2. The analysis of PoTable performance on varied task difficulty and table size is interesting. It would be further informative if the result breakdown on baseline methods are also provided, so it tells more clearly on which category the proposed PoTable method provides the most increases.

**Weaknesses:**

1. Unclear unique contribution of the proposed method: particularly, it is unclear to me where two main component highlighted are unique contributions of this paper.

(i) non-standard logical splits: (first of all, this term is somewhat vague so suggest changing it.) This paper proposed “standard” stages including (initialization, row selection, data type cleaning, reasoning, and final answering), where previous works such as Chain-of-Table (https://arxiv.org/abs/2401.04398) and Datar (https://arxiv.org/abs/2301.13808) have proposed similar decompositions, yet it is unclear why the split proposed in this paper is “more standard” than existing ones. Adding some clarifications on this could be helpful.

(ii) constrained operation pools: many table-related works do not have a restricted operation pool, for example, they can use arbitrary Python programs (ReAcTable: https://arxiv.org/abs/2310.00815, Chameleon: https://arxiv.org/abs/2304.09842) or even allowing model self-designed new operations (TroVE: https://arxiv.org/abs/2401.12869)

Overall, both dimensions claimed as novel in this work seem to be already explored by existing works. It would be helpful to highlight the differences of this work further to clarify the novelty aspect.
Further, it is not entirely clear to me why (i) and (ii) leads to human-like analysis, and what “human-like analysis” is defined/characterized in this work.

**Questions:**

1. How did you decide on the five steps as proposed in the PoTable method? And why do you believe that is the most "standard" split of the task? Can this split be generally applied to all table-reasoning tasks?

---

> ### Author Response · Authors · 2024-11-23
> **Official Response to Reviewer Xpqj**
>
> Thank you for your constructive feedback. We have carefully considered your suggestions and revised our manuscript. Here are our detailed responses to your concerns.
>
> > **W1.** Unclear unique contribution of the proposed method of non-standard logical splits and constrained operation pools. They need revision.
>
> We would apologize for our previous expression that have impacted your understanding of our study.
> This prompted us to rethink and refine the our motivation and contributions.
> Specifically, we fail to convey more essential concept of **integrating logical top-level guidance** in the first version of the manuscript.
> In this way, PoTable enjoys two advantages that make it a distinguished data analyst. **(1) Accuracy**: PoTable can easily plan coherent operation chains under precise and specific sub-goals with less possibility of
> misleading or missing steps, producing more accurate results. **(2) Explainability**: PoTable follows suitable structured top-level guidance with full operation execution, making it easier to verify the completeness and accuracy of the reasoning process.
>
> We sincerely thank you for your suggestions, which have significantly improved the motivation and contributions of our work.
>
> > **W2.** It is unclear why the stage split is standard, and it is unknown how this split can be generally applied to all table-reasoning tasks.
>
> As you concerned, we overemphasized that our stage split was **standard** in the first version of the manuscript.
> Our vague expression made you mistakenly believe that our division was the most standard, which was contrary to what we actually want to express.
> On the other hand, the logical stage split was in fact inspired by human data analysis and mining processes [1][2], aiming to make the reasoning results as accurate and explainable as possible.
> Therefore, in the revised version, we will describe the stage split as "relatively standard" because it indeed represents a widely-accepted top-level guidance among most researchers.
>
> As for general applications, our PoTable is actually extensive and scalable in some scenarios, yet needs a little customization in specific scenarios during implementation.
> This is because that all these stages are class instances of `PoTableBlock` in our implementation code.
> Specifically, for scenarios of multi-table and hierarchical tables, we may design a more sophisticated instructions for the initialization stage to store multiple tabel data frames (e.g., store them into `df1`, `df2` is a naive method) and tables with multi-level indexes (as work [3] designed as a distinct pre-processing stage), maybe with a more fine-grained stage split.
> As stated in the conclusion, we hope to enhance our approach in the future to better accommodate more advanced task scenarios.
>
> Thank you again for your suggestion, which inspires us to revise our manuscript in a more comprehensive manner.
>
> > [1] Fayyad U, Piatetsky-Shapiro G, Smyth P. The KDD process for extracting useful knowledge from volumes of data[J]. Communications of the ACM, 1996, 39(11): 27-34.
>
> > [2] Mariscal G, Marban O, Fernandez C. A survey of data mining and knowledge discovery process models and methodologies[J]. The Knowledge Engineering Review, 2010, 25(2): 137-166.

---

> > ### Comment · Reviewer_Xpqj · 2024-11-26
> >
> > I thank the authors for their additional explanation and willingness to improve the paper. Since we agreed on the listed concerns, I will keep my score.

---

> > > ### Author Response · Authors · 2024-11-27
> > > **Follow-up Responses to Reviewer Xpqj**
> > >
> > > Thank you for your feedback.
> > > In fact, we have already submitted a revised manuscript that addresses your concerns.
> > > We would like to learn more from your valuable suggestions towards our newly improved manuscript, so that we can make more progress to make this study better.
> > > Do you agree?

---

### Official Review · Reviewer_Hb7Z · 2024-10-30

**Soundness:** 2
**Presentation:** 3
**Contribution:** 1
**Rating:** 3
**Confidence:** 4

**Summary:**

This work proposes a pure prompt-based method to solve reasoning that involves table, namely PoTable. The author proposes to split the process explicitly into several stages: initialization, row selection, data type cleaning, reasoning and final answering. These stages include prompting API LLMs for planning, and for using pandas package to manipulate the table. The experiments show the proposed method performs better on certain types of table reasoning tasks.

**Strengths:**

See below

**Weaknesses:**

## Novelty

PoTable is a pure prompt-based method with hardwired reasoning stages designed specifically for solving a certain type of table reasoning tasks. On top of API LLMs, PoTable is also equipped with tool usage and self-correction, but both of these techniques are widely studied in prior work on various types of reasoning. The highlighted human-like reasoning stage design lacks justification and is somewhat overfitted to a certain type of table reasoning, leading to poor generalization over other types of questions. That said, this work lacks novelty.


## Quality
Several methodological concerns undermine the paper's quality:

Hardwired reasoning stages:
- The authors claim that the 5-stage split represents "standard logic" (L85, L90), but there no direct evidence is provided to justify why these particular stages should be considered standard.
- Scalability issue: How flexible is this setup? Though it is claimed to be standard, can this approach handle general tabular reasoning scenarios? How does it perform with complex queries requiring merging tables? What about multi-step reasoning problems?

Pipeline design choices not well-motivated or -justified:
- Why only use few-shot examples at the final stage?


## Clarity

This paper is generally easy to follow.


## Significance

As mentioned above, this work targets a specific type of table reasoning tasks that involve single-step, single-table reasoning. While being able to beat the baseline methods, the performance gain is moderate. More importantly, the introduced 5-stage reasoning significantly limits its application to other types of reasoning, further limits its impact on the field. That said, the significance is minor.

**Questions:**

See above

---

> ### Author Response · Authors · 2024-11-23
> **Official Response to Reviewer Hb7Z**
>
> Thank you for your constructive feedback. We have carefully considered your suggestions and revised our manuscript. Here are our detailed responses to your concerns.
>
> > **W1.** This work lacks novelty, since the highlighted human-like reasoning stage design lacks justification.
>
> We would apologize for our previous expression that have impacted your understanding of our study.
> This prompted us to rethink and refine the our motivation and contributions.
> Specifically, we overemphasized that our stage split was **standard** rather than conveying more essential concept of **integrating logical top-level guidance**.
> In this way, PoTable enjoys two advantages that make it a distinguished data analyst. **(1) Accuracy**: PoTable can easily plan coherent operation chains under precise and specific sub-goals with less possibility of
> misleading or missing steps, producing more accurate results. **(2) Explainability**: PoTable follows suitable structured top-level guidance with full operation execution, making it easier to verify the completeness and accuracy of the reasoning process.
>
> Furthermore, the logical stage split was inspired by human data analysis and mining processes [1][2], aiming to make the reasoning results as accurate and explainable as possible.
> In the revised version, we will describe the stage split as "relatively standard" because it indeed represents a widely-accepted top-level guidance among most researchers.
> We sincerely thank you for your suggestions, which have significantly improved the motivation and contributions of our work.
>
> > **W2.** Pipeline design choices not well-motivated or -justified, since it is unknown why only use few-shot examples at the final stage.
>
> Our framework leveraged two capabilities of the LLM: operation planning and code generation.
> In the planning phase, we use few-shot examples across all stages since they have clear and precise sub-goals.
> As for code generation, however, we need to generate appropriate code for each operation.
> During this process, we don't know the specific content of each fine-grained operation in advance, but only know that it is a crucial component to accomplish the stage goal.
> Therefore, we cannot provide any examples for few-shot learning.
> The final stage's output for the answers is consistent across all samples in the specific task, and we must ensure that this stage can generate executable code with the `print()` method to produce valid outputs. Hence, we use few-shot examples for this stage.
>
> In other words, your mention of **only use few-shot examples at the final stage** means "**during code generation**, we only use few-shot examples at the final stage", since in the planning phase we always use few-shot examples for better planning.
>
> > **W3.** This method is somewhat overfitted to a certain type of table reasoning, leading to poor generalization over other types of questions (e.g., complex queries, merging, multi-step).
>
> Our PoTable is actually extensive and scalable, yet needs a little customization in specific scenarios during implementation.
> This is because that all these stages are class instances of `PoTableBlock` in our implementation code.
> Specifically, for scenarios of multi-table and hierarchical tables, we may design a more sophisticated instructions for the initialization stage to store multiple tabel data frames (e.g., store them into `df1`, `df2` is a naive method) and tables with multi-level indexes (as work [3] designed as a distinct pre-processing stage), maybe with a more fine-grained stage split.
> Regarding the scenarios you mentioned (e.g., complex queries, merging, multi-step processes), our current method requires further optimization to address these more intricate task scenarios.
> The core contribution of this paper does not focus on these high-level tasks.
> As stated in the conclusion, we hope to enhance our approach in the future to better accommodate more advanced task scenarios.
> Thank you again for your suggestion, which inspires us to revise our manuscript in a more comprehensive manner.
>
> > [1] Fayyad U, Piatetsky-Shapiro G, Smyth P. The KDD process for extracting useful knowledge from volumes of data[J]. Communications of the ACM, 1996, 39(11): 27-34.
>
> > [2] Mariscal G, Marban O, Fernandez C. A survey of data mining and knowledge discovery process models and methodologies[J]. The Knowledge Engineering Review, 2010, 25(2): 137-166.
>
> > [3] Cao Y, Chen S, Liu R, et al. API-Assisted Code Generation for Question Answering on Varied Table Structures[C]. Proceedings of the 2023 Conference on Empirical Methods in Natural Language Processing. 2023: 14536-14548.

---

> > ### Comment · Reviewer_Hb7Z · 2024-12-02
> > **Response**
> >
> > Thanks for the response. I think my concern is less about the overemphasis on the term "standard" but the novelty and scalability of the proposed approach as a whole: this is a prompt-based method with predefined stages for table reasoning. The fact that you need to redesign the prompt and stages for different types of table reasoning problems indicates that this method cannot scale without human intervention. Given that there are many strong general multistep reasoners available (openai o1, deepseek QwQ), one needs a good reason to justify the need for such a pipeline: is this task so important and special that it requires a dedicated pipeline? Is it so difficult that existing general reasoners cannot solve it properly? These are things that need to be addressed before it can be accepted. That said, I'm keeping my score.

---

### Official Review · Reviewer_NPzY · 2024-11-04

**Soundness:** 2
**Presentation:** 3
**Contribution:** 2
**Rating:** 5
**Confidence:** 3

**Summary:**

This paper introduces POTABLE, a table-based reasoning method that simulates human analyst behavior through a staged analytical approach. The method combines a Python interpreter for execution with an LLM-based operation planner and code generator. The method mimics human-like logical stage split (initialization, row selection, data cleaning, reasoning, and answering) and open-world operation space. The authors demonstrate POTABLE's effectiveness through experiments on WikiTQ and TabFact datasets, showing performance improvements.

**Strengths:**

1. The paper provides a systematic refinement of existing table-based agents. By breaking down the process into well-defined stages and implementing careful error handling, POTABLE addresses common failure modes of general-purpose agents. The empirical results demonstrate that these refinements lead to meaningful performance improvements, showing the value of structured approaches to agent design.

2. The authors test their method across different datasets (WikiTQ and TabFact), model backbones (GPT and LLAMA), and conduct detailed ablation studies. The performance improvement over baselines (>4% on multiple datasets) suggests that the structured approach does offer tangible benefits over more generic agent implementations.

**Weaknesses:**

1. The idea of using LLMs with pandas for table reasoning has already been previously explored in various works. For instance, LangChain has implemented pandas agents [1] at least a year ago, and at least 2 paper [2][3] (and I think there are more as this is a paper one year ago) has discussed the relationship between agent-based and direct reasoning approaches for tabular understanding. Under this situation, the main contribution will be in providing a more refined (and maybe fixed) workflow for table reasoning, which might have diminishing returns with more capable models.
2. The paper lacks a crucial analysis of token efficiency and alternative approaches. A key consideration for any agent-based system is whether the increased token consumption from multiple LLM calls is justified by the performance gains. For instance, an alternative approach using self-consistency with multiple single-pass attempts might achieve similar accuracy while being more token-efficient. Without this comparison, it's difficult to determine if the improved accuracy (>4%) justifies what could be a significant increase in token usage. The paper should have included a detailed analysis comparing token consumption with simpler approaches and demonstrating why their multi-stage workflow is more effective than alternative methods like self-consistency.
3. There are several fundamental limitations in current (especially programming-based) table agents. Specifically, these agents (including POTABLE) struggle with three key scenarios: (1) multi-table reasoning that requires joining or comparing information across tables, (2) hybrid reasoning that combines tabular data with textual context, and (3) handling irregular table structures with multiple headers or nested hierarchies. While these challenges may be outside the paper's intended scope, they represent critical real-world use cases that limit the practical applicability of such methods.

---
[1] https://python.langchain.com/docs/integrations/tools/pandas/

[2] "Rethinking Tabular Data Understanding with Large Language Models", NAACL 2024, https://arxiv.org/abs/2312.16702

[3] "API-Assisted Code Generation for Question Answering on Varied Table Structures", EMNLP 2023, https://arxiv.org/pdf/2310.14687

**Questions:**

- See weakness and:
- How do you envision POTABLE could be extended to handle multi-table scenarios or hybrid reasoning with both tabular and textual data or irregular table structures ?
- As LLM capabilities continue to advance, would you agree or maybe disagree that the benefits of structured workflows might need to be reconsidered? (One could argue that giving more autonomy to stronger models - perhaps with fewer constraints on their reasoning process - might actually lead to better performance, but i would be interested in your thoughts on this trade-off between structure and autonomy.)

---

> ### Author Response · Authors · 2024-11-23
> **Official Response to Reviewer NPzY [1/2]**
>
> Thank you for your constructive feedback. We have carefully considered your suggestions and revised our manuscript. Here are our detailed responses to your concerns.
>
> > **W1.** The idea of using LLMs with pandas for table reasoning has already been previously explored in various works. The main contribution will be in providing a more refined (and maybe fixed) workflow for table reasoning, which might have diminishing returns with more capable models.
>
> As you mentioned, using LLMs with pandas is not the core innovation of this study. Instead, inspired by the stage-split principles that a distinguished tabular data analyst may follow, our study inject logical top-level guidance into the tool-based reasoning framework by spliting several distinct logical stages. In this way, our PoTable can easily plan coherent operation chains under precise and specific sub-goals, and the completeness and accuracy of the whole reasoning process can be easily validated.
>
> Although the improvement may decline as the capabilities of the base LLM increase, there is a rising demand of numerous domain-specific reasoning scenarios rather than general reasoning, which might need more specific intrinsic guidance or knowledge.
> Compared with feeding LLMs with massive data from different domains, similar top-level guiding thinking may greatly improve the reasoning effect in a faster and more economical way.
> This is also a major direction we will explore in the future.
> Thanks again for your inspirations to us.
>
> > **W2.** The paper lacks a crucial analysis of token efficiency and alternative approaches.
>
> Many thanks for your valuable suggestion, which promotes us to conduct efficiency experiments on TabFact (S) for GPT-based PoTable and other three representative baselines. Please check the results in Table 1.
>
> **Table 1.** Efficiency experiments on TabFact (S) for GPT-based methods.
> For *Binder*, *Dater* and *Chain-of-Table*, we compared the results of single LLM generation (**n=1**, explained in *Appendix A*) and default LLM generation (**Default**, which means `n_samples` follows their claimed settings in the corresponding article).
> In addition, we estimate the LLM generation counts of each method under default setting.
> |Approach|Accuracy (n=1 / Default)|# Generation (Default)|Details|
> |:-:|:-:|:-:|:--|
> |Binder|84.63 / 85.13|50|Generate SQL: 50|
> |Dater|80.98 / 82.26|100|Generate Decomposition: 40, Generate Cloze: 20, Generate SQL: 20, Query: 20|
> |Chain-of-Table|84.24 / 85.23|<=22|Dynamic Planning: <=4 (3.74 on average), Generate Args: <=17 (16.09 on average), Query: 1|
> |**PoTable (*only* Reason)**|85.92|<=6|Planning: 1, Code Generation: <=4 (3.72 on average), Regeneration: <=1 (less than 1 on average)|
> |**PoTable**|**88.93**|<=10|Planning: 3, Code Generation: <=6 (5.60 on average), Regeneration: <=1 (less than 1 on average)|
>
> Based on the result in Table 1, we make the following analysis.
>
> - We conducted experiments of baselines under **Default** settings, and noticed that they did achieve some improvement compared to results of **n=1** settings. However, PoTable always follows the **n=1** setting yet outperforms these baselines, which fully demonstrates the effectiveness of PoTable.
>
> - On the other hand, we compared the LLM generation counts to evaluate the efficiency of these methods. In *Binder* and *Dater*, the generation counts are fixed as 50 and 100 respectively, while the ones of *Chain-of-Table* and our *PoTable* are fluctuated dynamically. Therefore, we calculated the empirical average counts of each module and rounded them up as their estimation (shown in **Details**). It can be seen that PoTable has much fewer LLM generation counts compared to previous baselines.
>
> - In addition, the LLM generation count increase of PoTable than PoTable (*only Reason*) is limited, while the accuracy improvement is more than 3%. Among them, the logical stage split slightly increases the LLM generation counts of *Operation Planning* and *Code Generation*, but under clearer stage guidance, the planning module is less likely to produce wrong plans or overlook important details. Such results validate the efficiency and effectiveness of our PoTable.
>
> In summary, the effectiveness of our method comes from the design of logical stage planning and code execution rather than multiple LLM generation.
> This part of the analysis results will be updated to the next submission as the optimization part of our work.
> Thank you again for your valuable suggestion that inspires our study to be better improved.

---

> ### Author Response · Authors · 2024-11-23
> **Official Response to Reviewer NPzY [2/2]**
>
> > **W3.** Programming-based table agents may struggle with three key scenarios: multi-table reasoning, text-table hybrid reasoning and irregular structured table reasoning (*e.g.*, hierarchical tables).
>
> This question is essentially the same as Q1, namely:
>
> > **Q1.** How do you envision PoTable could be extended to handle multi-table scenarios or hybrid reasoning with both tabular and textual data or irregular table structures?
>
> Our PoTable is actually extensive and scalable, yet needs a little customization in specific scenarios.
> This is because that all these stages are class instances of `PoTableBlock` in our implementation code.
> Specifically, for multi-table and hierarchical tables, we may design a more sophisticated instructions for the initialization stage to store multiple tabel data frames (e.g., store them into `df1`, `df2` is a naive method) and tables with multi-level indexes (as work [1] designed as a distinct pre-processing stage), maybe with a more fine-grained stage split.
> In text-table hybrid reasoning, as our reasoning results are generated by full program execution, the current PoTable cannot handle it directly.
> Therefore, another final LLM-query step is required as an augmentation to summarize its comprehension and the executed result to draw the final conclusion.
>
> > **Q2.** As LLM capabilities continue to advance, would you agree or maybe disagree that the benefits of structured workflows might need to be reconsidered?
>
> With the improvement of capable LLMs, numerous studies no longer focus on structured design but autonomous exploration to stimulate the potential LLMs.
> Despite their success, the completely autonomous exploration method is prone to losing some details or planning some redundant operations in some complex situation reasoning scenarios, and it is difficult to obtain reliable results in some specific field scenarios because it is time-consuming for verification.
> Consequently, we hope to inject logical top-level guidance in a suitable manner into our reasoning framework to achieve a balance between structure and autonomy.
> The experimental results also prove the effectiveness of this idea (as can be seen from the comparison in Table 1).
> In the future, in more complex and special scenarios, we hope to continue to explore more effective balance methods so that the correctness and explainability of the reasoning results can be well guaranteed.
> We would like to thank you again for raising this issue, which promotes us to optimize the expression of our core motivation and contribution of this study.
>
> > [1] Cao Y, Chen S, Liu R, et al. API-Assisted Code Generation for Question Answering on Varied Table Structures[C]. Proceedings of the 2023 Conference on Empirical Methods in Natural Language Processing. 2023: 14536-14548.

---

### Author Response · Authors · 2024-11-23
**Overall Revision of the Submitted Manuscript**

We would like to thank all the reviewers for their efforts in reviewing our manuscript.
Your valuable suggestions inspire us to improve our study and revise our manuscript in a graceful manner.
Specifically, we have resubmitted a newly revised manuscript following your suggestions.
The main revision parts can be summarized as below.

**1. Motivation and Contribution**

We apologize for our previous expression that has impacted your understanding of our study.
This prompted us to rethink and refine our motivation and contributions.
Specifically, we overemphasized that our stage split was **standard** rather than conveying a more essential concept of **integrating logical top-level guidance**, as some existing reasoning approaches emphasize extensive and flexible utilization of symbolic tools without fully considering the intrinsic logic of the overall reasoning process.
Therefore, we have revised the corresponding expressions about motivation and contribution in the abstract, introduction and contribution.
Some related expressions (*e.g.*, advantages) have also been improved, without modifying our implemented approach.

**2. Experiments**

We have conducted efficiency experiments and added a new related subsection into the experimental part with results and analysis.
In addition, more details in the comparison experiments and ablation studies have also been improved.

Apart from the above, we have improved the overall presentation of our new manuscript.

**3. Title Revision on System**

According to the author guide in https://iclr.cc/Conferences/2025/AuthorGuide, we are allowed to revise the title of our manuscript, yet the revision system does not provide a block for title selection. Therefore, please check our new title in the revised manuscript as **POTABLE: PROGRAMMING ON TABLES TO REASON LIKE A DISTINGUISHED HUMAN DATA ANALYST**.

Anyway, thank you again for your valuable suggestions. If you have any other concerns, feel free to contact us.

Sincerely,

Paper 10382 Authors

---

### Meta-Review · Area_Chair_GqoT · 2024-12-25

**Metareview:**

This paper concerns table-based reasoning with LLMs and proposes PoTable, which combines an LLM-based operation planner and code generator with a python interpreter for real-time execution. PoTable mimics a human-like logical stage split and expanding the operation pool into an open-world space. Experimental evaluations show that PoTable outperforms existing approaches on two public benchmarks (i.e., WikiTQ and TabFact) by over 4% absolute accuracy improvement. The idea of using LLMs for table reasoning has been extensively explored by many recent works (references shared by reviewers, NPzY and Xpqj) and the proposed method follows a similar (or refined) workflow, making the proposed method incremental. Furthermore, the new method relies on multiple LLM calls and has a low token efficiency compared to existing approaches. Comparing the performance without analyzing the computation costs (in terms of LLM calls)  making the 4% improvement sound less promising.

**Additional Comments On Reviewer Discussion:**

During the rebuttal, the authors and reviewers mainly discussed the novelty and effectiveness compared to the previous works. The authors carefully elaborated the main differences. The authors and reviewer Hb7Z have significant disagreements regarding the key novelty and contributions, making the discussion not converged to a reasonable consensus. Overall, reviewers remain less convinced about the significant novelty in contrast to the previous work.

---

### Decision · Program_Chairs · 2025-01-22

Reject